# Detection and Molecular Characterization of Enteric Viruses in Poultry Flocks in Hebei Province, China

**DOI:** 10.3390/ani12202873

**Published:** 2022-10-21

**Authors:** Libao Chen, Ligong Chen, Xuejing Wang, Shuying Huo, Yurong Li

**Affiliations:** 1College of Veterinary Medicine, Hebei Agricultural University, Veterinary Biological Technology Innovation Center of Hebei Province, Baoding 071001, China; 2Institute of Animal Husbandry and Veterinary Medicine of Hebei Province, Baoding 071000, China

**Keywords:** chicken, enteric virus, recombination, diversity, phylogenetics

## Abstract

**Simple Summary:**

Enteric viruses act as etiological agents for a series of health disturbances that pose a threat to commercial chickens worldwide. The affected chickens exhibit stunted growth, low feed conversion, etc. On a global scale, research on enteric virus diversity has been performed in countries such as India, South Korea and Brazil, yet at present, there have been no conclusive reports of avian enteric viruses in China. In the present study, the virus species, infection types, clinical symptoms, and relationships among the virus species were studied in 145 positive enteric virus samples. Additionally, the evolutionary relationship and recombination of the viruses were also further studied. The results of this study can be used to define the distribution and infection type of enteric viruses in poultry, and to analyze the classification of and evolutionary relationship between certain viruses.

**Abstract:**

Enteric viruses, as a potential pathogen, have been found to be vital causes of economic losses in poultry industry worldwide. The enteric viruses widely studied to date mainly include avian nephritis virus (ANV), avian reovirus (ARe), chicken astrovirus (CAstV), chicken parvovirus (ChPV), fowl adenovirus group I (FAdV-1), infectious bronchitis virus (IBV), and avian rotavirus (ARoV). This paper aimed to identify single and multiple infections of the seven enteric viruses using the data obtained from positive 145 enteric virus samples in poultry flocks from different areas in Hebei Province, throughout the period from 2019 to 2021. Next, the correlation between bird age and clinical signs was investigated using PCR and RT-PCR techniques. Furthermore, the whole genomes of seven parvovirus strains and open reading frame 2 (ORF2) of six CAstV strains and eight ANV strains were sequenced for phylogenetic analysis and recombination analysis, to characterize the viruses and evaluate species correlation and geographic patterns. A total of 11 profiles of virus combinations were detected; 191 viruses were detected in 145 samples; 106 single infections were reported in 73.1% of the samples; and multiple infections were detected in the remaining 26.9%. For viruses, 69% of ChPV was correlated with single infection, while ANV (61.4%) and CAstV (56.1%) were correlated with multiple infections. However, IBV and ARe were not detected in any of the samples. Recombination events were reported in parvovirus, and all CAstV sequences investigated in this paper were included within genotype Bii. The eight ANV strains pertained to different subtypes with significant differences. The above results revealed for the first time the complexity of enteric viruses over the past several years, thus contributing to disease prevention and control in the future.

## 1. Introduction

Enteric viruses act as the etiological agents for a series of health disturbances that pose a threat to commercial chickens worldwide. Diseases related to enteric viruses were first reported in the late 1970s. These diseases are characterized by growth deficiency, pale bird syndrome, helicopter wing disease, and other abnormalities [1,2,3,4]. Although these enteric diseases have proven very difficult to reproduce experimentally with defined viral inoculums [5], their hazards are real and are mainly associated with low feed conversion rates, decreased immune function, and high mortality [2,3,5]. A considerable number of viruses have been detected in or isolated from the intestinal tract of poultry, whereas the main enteric viruses reported that caused poultry enteric diseases consist of chicken parvovirus (ChPV), fowl adenovirus of group-I (FAdV-I), infectious bronchitis virus (IBV), two viruses of the *Astroviridae* family: avian nephritis virus (ANV) and chicken astrovirus (CAstV), and two viruses of the *Reoviridae* family: avian reovirus (ARe) and avian rotavirus (ARoV) [4,6,7,8]. The results of molecular epidemiological investigation of different countries and regions suggested that different viruses or virus combinations have existed [8,9]. In Brazilian poultry flocks, single infections outnumbered multiple infections, and the main enteric virus was IBV. In South Korea, however, multiple infections were identified in 51.7% of chicken farms, and the main enteric virus was ANV. In China, the first report of enteric virus infections in chickens was published in 1987. Chicken flocks showed growth inhibition, and the CHPV was isolated. Since then, there have been reports about duck astrovirus, CAstV, ANV, and FAdV [10,11,12,13]. For the epidemiological investigation of enteric virus, most of the reports are about a specific single virus in the chicken population [11,12,14]. Conventional PCR and reverse transcription-PCR (RT-PCR) have been recognized as the two most commonly used methods to perform molecular epidemiological investigation of viruses in the poultry industry [8]. For different enteric viruses, the possible origin and genetic evolution trend of the virus isolates have been unknown in Hebei Province. This paper aimed to determine the prevalence of enteric viruses affecting commercial chicken flocks in Hebei Province, which included an analysis of the correlations between single and multiple infections, bird age, and clinical signs. We carried out PCR amplification of complete genome sequences of seven parvovirus strains, eight ANV open reading frame 2 (ORF2) genes, and six CAstV ORF2 genes sequenced from parts of Hebei isolates, to gain insight into molecular evolutionary characteristics and taxonomic status.

## 2. Materials and Methods

### 2.1. Field Samples

In this study, 145 positive samples of single and multiple enteric viral infections were detected from 212 samples (185 pullet/layer Hens and 27 broilers) collected from large-scale farms of commercial chickens in Baoding, Cangzhou, Hengshui, Shijiazhuang, Xingtai, Zhangjiakou, and Langfang of Hebei Province, China (Appendix A). Among the positive samples, 21 samples were collected from broilers of 2–5-week-olds, and 124 from pullet and layer hen flocks of 2–49-week-olds. Sampling was performed from 2019 to 2021. Each sample consisted of a pool of maximum three small intestine samples, and each sample was taken from different farms throughout the abovementioned cities. The main reported symptoms of the birds were digestive problems (including enteritis, diarrhea, decreased feed absorption, and indigestion of feed), signs of respiratory disease, and stunting syndrome.

### 2.2. Preparation of Samples

The samples were taken from the small intestine. A 1:4 saline was prepared for grinding; freeze-thaw was repeated 3 times; the cell debris was removed by centrifuging at 8000 RPM for 10 min. The supernatants were filtered through a 0.22 µm filter to remove bacteria, and stored at −80 °C in preparation for the extraction of viral DNA and RNA.

The samples were processed and analyzed by conventional PCR and RT-PCR for detection of the seven main enteric viruses reported in recent years: FAdV-I, ChPV, CAstV, ANV, IBV, ARe, and ARoV. The organ used for viral detection was the intestines. The results were categorized as previously described [7], mainly according to the viral infection in each sample, the type of birds (broilers, layers), the age of birds (days for broilers and weeks for pullet/layer hen), and clinical signs. Known positive samples for FAdV-1, ChPV, CAstV, ANV, and ARoV were validated through Sanger sequencing, and the corresponding positive control viruses were provided by the Disease Testing Center Laboratory of Hebei Agricultural University.

### 2.3. Viral Genome Extraction and Reverse Transcription 

The viral genome was extracted using an EasyPure Viral DNA/RNA Kit (TransGen Biotech, Beijing, China). Reverse transcription was performed by Superscript SMART MMLV Reverse Transcriptase (Takara, Beijing, China) with random primer (Takara, Beijing, China), according to the manufacturer’s instructions.

### 2.4. PCR Detection of Viruses

PCR was performed by a previously described method to screen all samples for the presence of the seven tested virus genes of ANV, CAstV, ARoV, ARe, IBV, FAdV-I, and ChPV [9,15,16,17,18]. The primers used for pentaplex PCR are listed in Table 1. PCR amplification was performed using 2 μL of the template DNA (ChPV and FAdV-I) or cDNA (the other five viruses), 2 μL of each primer, dNTP Mixture (2.5 mM) 4 μL, 10 × PCR Buffer (Mg^2+^ plus) 5 μL, RNase-free water 34.5 μL, and 0.5 μL of Taq DNA polymerase, for a total volume of 50 μL. The following temperature conditions were used for the PCR reaction: 1 cycle at 94 °C for 3 min, 35 cycles of amplification (94 °C for 30 s, annealing treatment for 45 s, 72 °C for 1 min), and a final extension at 72 °C for 10 min.

### 2.5. PCR Amplification of Complete Genome Sequences of Parvovirus

PCR was performed following a previously described method [6,14]. The primers used for pentaplex PCR are listed in Table 2. PCR amplification was performed using 2 μL of the template DNA. The same procedure described above was used, except for the annealing temperature and extension time. The details are listed in Table 2.

### 2.6. PCR Amplification of Complete ORF2 for CAstV and ANV

PCR was used to amplify ORF2 complete sequences of ANV [19] and CAstV. The primers used for pentaplex PCR are listed in Table 3. PCR amplification was performed using 2 μL of the cDNA. The same procedure described above was used, except for annealing temperature and extension time. The details are listed in Table 3.

### 2.7. Evaluation of RT-PCR and PCR Products

Next, 5 μL of each PCR product was electrophoretically separated and visualized by ethidium bromide staining. Some PCR products were sent for sequencing, then compared to the information in the GenBank database of the BLAST network of the National Center for Biotechnology Information.

### 2.8. Sequencing and Phylogenetic Analysis

Some samples were chosen randomly for DNA sequencing to validate the RT-PCR and PCR-specific reactions to screens for the enteric viruses: ChPV (*n* = 9, MW548986–MW548989, MZ546785–MZ546789), ANV (*n* = 4, MW522990–MW522993), CAstV (*n* = 1, MW855893), and FAdV-I (*n* = 1, MW855894).

PCR production of the complete sequence of parvovirus and ORF2 complete sequence of ANV and CAstV were chosen randomly for DNA sequencing to be used in the phylogenetic analysis: ChPV (*n* = 7): isolate ChPV/CK/CH/HB/17 (OK274225), isolate ChPV/CK/CH/HB/31 (OK274226), isolate ChPV/CK/CH/HB/52 (OK274227), isolate ChPV/CK/CH/HB/77 (OK274228), isolate ChPV/CK/CH/HB/81 (OK274229), isolate ChPV/CK/CH/HB/82 (OK274230) and isolate TuPV/CK/CH/HB/22 (OK490349); ANV (*n* = 8): isolate ANV/CK/CH/HBBD/32 (OK274237), isolate ANV/CK/CH/HBCZ/69 (OK274238), isolate ANV/CK/CH/HBBD/77 (OK274239), isolate ANV/CK/CH/HBCZ/106 (OK274240), isolate ANV/CK/CH/HECZ/117 (OK274241), isolate ANV/CK/CH/HESJZ/126 (OK274242), isolate ANV/CK/CH/HESJZ/127 (OK274243), and isolate ANV/CK/CH/HBBD/134 (OK274243); CAstV (*n* = 6): isolate CAstV/CK/CH/HBBD/6 (OK274231), isolate CAstV/CK/CH/HBCZ/26 (OK274232), isolate CAstV/CK/CH/HBLF/41 (OK274233), isolate CAstV/CK/CH/HBBD/52 (OK274234), isolate CAstV/CK/CH/HBSJZ/80 (OK274235), and isolate CAstV/CK/CH/HBCZ/117 (OK274236). All sequences had already been submitted to the GenBank database. Phylogenetic trees were inferred using the neighbor-joining statistical method, with 1000 bootstrap replications in the software package of the Molecular Evolutionary Genetics Analysis 11 software program (Pennsylvania State University, State College, PA, USA). 

### 2.9. Recombination Analysis

The sequence analysis was performed using RDP4 software v5.5, and seven recombination and analysis methods were used, as described by Palomino-Tapia et al. (2020) [20]. The putative recombination sequences were further investigated using Simplot analysis within the Simplot program v3.5.1, with the following parameters: window size = 400 bp; step size = 40 bp; GapStrip = On; Kimura 2-parameter substitution model; T/t = 2; and using the neighbor-joining method.

### 2.10. Statistical Analysis

Descriptive statistics were used to represent the variability of the positive samples as previously described [7], including single and multiple viral infections, type of birds (broilers and pullet/layers), age of birds (days for broilers and weeks for pullet/layers hen), and clinical signs (respiratory signs, digestive signs, stunting, and no clinical signs reported). Next, statistical analyses were performed using the SPSS software package version 19.0 (SPSS Inc., Chicago, IL, USA). Categorical variables were reported as absolute, along with the corresponding relative frequencies. Chi-squared or Fisher’s exact test was used for comparisons. The two-sided significance level alpha was set as 0.05.

## 3. Results

### 3.1. Single and Multiple Viral Infections

A total of 191 viruses were found in 145 positive samples, and the most common virus detected corresponded to ChPV, showing a 52.4% (100/191) occurrence, followed by ANV, with 23.0% (44/191); CAstV, with 21.5% (41/191); FAdV-I, with 2.6% (5/191), and ARoV, with 0.5% (1/191) (Table 4). However, IBV and ARe were not detected in the samples. Single infections were found in 73.1% (106/145) of the samples, with ChPV being the most predominant virus diagnosed as a unique agent. In addition, ChPV, ANV, and CAstV were found in simple and combined (2 to 4 viruses) infections. FAdV-I was found in unique and combined (two to three viruses) infections. ARoV was only found in combined (four viruses) infections. The multiple infection analysis results revealed that two viral infections were found in 22.8% (33/145) of samples, followed by three viral infections in 3.4% (5/145), and four viral infections in 0.7% (1/145). The single and multiple viral infections are described in Table 4. 

The analysis of the factors of 191 virus distribution suggested that the ChPV in single and multiple infection respectively accounted for 69.0% and 31.0%, while those of ANV were 38.6% and 61.4%, and those of CAstV were 43.9% and 56.1%, respectively (Table 4). Chi-squared and Fisher’s exact tests were performed in terms of the distribution enteric viruses between two types of infection (single and multiple), in accordance with the 191 viruses detected. There were extremely significant differences in the distribution of ChPV, ANV, and CAstV between single and multiple infections (Table 4). This revealed that ChPV was more common in single infections, while ANV and CAstV were more common in multiple infections. 

Infections occurred with 11 profile combinations of viruses (Table 5). ChPV was present in combination with ANV, FAdV-I, and ARoV, resulting in two and three virus combinations. ARoV was only present in combination with ChPV, ANV, and CAstV, resulting in four virus combinations. In addition, ChPV, ANV, CAstV, and FAdV-I were all present within a single infection. 

### 3.2. Age and Type of Birds

A total of 185 samples received for viral diagnostics originated from broilers older than 15 days and pullet/layer hens older than 2 weeks (Appendix A). In accordance with the age of the birds, the samples corresponding to pullet/layer hens were assigned to five groups, with the intervals of 5 weeks of age, from week 2 to week 26, and one additional group for birds older than 26 weeks (Table 6). The samples corresponding to broilers were assigned to three groups with the intervals of 7 days of age, from day 15 to day 35 (Table 6). The positive samples from pullet/layer hens in the 2–6-week groups were found to have the highest rate of viral infections, which was 87.2%. The viral infection rates of the other pullet/layer hen groups were measured as 67.4% for the 7–11-week group, 82.4% for the 12–16-week group, 65.9% for the 17–21-week group, 50.0% for 22–26-week group, and 29.4% for the older than 26 weeks group (Table 6). Broilers were found to have the highest rate of viral infections at 100% for the 15–21-day group. The viral infection rate in the 22–28-day and 29–35-day groups were found to be 75.0% (Table 6). 

FAdV-I was detected in the samples from broilers (1/5) and pullet/layer hens (4/5). ChPV was detected in the samples from broilers (18/100) and pullet/layer hens (82/100). CAstV was reported in the samples from pullet/layer hens (36/41) and broilers (5/41). ANV was found in the samples from broilers (10/44) and pullet/layer hens (34/44). ARoV was reported only in one pullet/layer hen sample, whereas ARe and IBV were not detected in any chicken line. All data describing the positive samples in accordance with the age of layers and broilers affected by the enteric viruses in this paper are listed in Table 7.

### 3.3. Clinical Signs

Single viral infections were detected primarily in pullet/layer hens, at 97/124 (78.22%), followed by 9/21 (42.86%) in broilers. Multiple viral infections were reported with a higher frequency in broilers, at 11/21 (52.38%) with two viruses and 1/21 (4.76%) with three viruses. The only sample with four viral infections was found in the pullet/layer hens (Table 8). A chi-squared test and Fisher’s exact test were performed between two chicken lines in accordance with 145 samples detected. There were extremely significant differences (*p* < 0.01) both in single infection and dual virus infection between the broilers and pullet/layer hens (Table 4). In the broilers, dual virus was found as the main infection type, while in the pullet/layer hens, single infection was found as the main infection type.

The positive samples fell into four groups according to the clinical signs and type of birds. The most common disease symptoms reported were digestive problems in 54/145; respiratory signs were detected in 16/145 samples; stunting signs were detected in 33/145; and there were no reported signs in 42/145. The broilers and pullet/layer hens showed the highest values of digestive problems. The digestive problems included were defined as intestinal inflammation, indigestion of feed, and watery stool. Table 8 lists all of the frequencies of clinical signs. The results of the Fisher’s exact test suggested that there was no significant difference between the clinical signs for the respective type of birds (*p* > 0.05).

### 3.4. Phylogenetic Analysis and Recombination Analysis of Parvovirus

The seven isolates corresponding to parvovirus were studied phylogenetically with strains from different countries based on the complete sequence gene (Figure 1). The 28 reference strains (Appendix A) could be assigned to two branches, namely ChPV and TuPV, while the parvoviruses of the seven Hebei strains were all clustered with ChPV. Isolate CHPV/CK/CH/HB/81 was clustered with the reference sequence of strain GX-CH-PV-23, and isolate CHPV/CK/CH/HB/82 was found to have the closest genetic distance from the reference strain GX-CH-PV-15. The above two strains were located in a large clade, and clustered further with the strains from South Korea and the US. Additionally, isolates ChPV/CK/CH/HB/52, 31, and 17 were clustered in a small group, which was closest to the reference strain 260 from the US and ABU-P1 from Hungary; isolate ChPV/CK/CH/HB/77 was clustered with the reference sequence of strain GA/1472/2004 from the US; and isolate TuPV/CK/CH/HB/22 stood alone in a cluster. 

The phylogenetic tree of the complete parvovirus sequences gene (Figure 1) was clustered in a similar manner as the phylogenetic tree based on the full coding sequence of the VP1 (Appendix A), whereas the above two phylogenetic trees were clustered differently as the phylogenetic trees of NS (Appendix A). Isolate TuPV/CK/CH/HB/22 was clustered with the reference sequences in the group of TuPV, and a second group of isolates (ChPV/CK/CH/HB/81, 82, 77, 52, 31, and 17) were clustered with the reference sequences of ChPV (Appendix A). As revealed by the phylogenetic trees, isolate TuPV/CK/CH/HB/22 may have undergone recombination events.

Next, the putative recombinant sequences isolate TuPV/CK/CH/HB/22 were investigated with seven recombination methods in RDP4, followed by SimPlot analysis using the Simplot program. The results of the above analyses suggested that there was a recombination event of TuPV/CK/CH/HB/22 strain, with TuPV-1090 as the major parent and ChPV/CK/CH/HB/31 as the minor parent, and with the *p*-Value range of 1.274 × 10^−12^–7.748 × 10^−96^. The recombination region was located in the NS gene and part of the VP1 gene, and the recombination site was located between 287 and 3234 bp (Figure 2). As confirmed by the recombinant analysis, TuPV/CK/CH/HB/22 was a recombinant strain.

### 3.5. Phylogenetic Analysis of CAstV and ANV

The six isolates corresponding to CAstV were investigated phylogenetically with 13 strains (Appendix A) from different countries, in accordance with the complete sequence gene of the ORF2 (Figure 3). PCR products and corresponding nt sequences (2232 bp) of a complete region gene of CAstV were obtained through RT-PCR. Among the six CAstV identified in this paper, the nt and predicted amino acid sequences of the CAstV ORF2 gene were found to exhibit high sequence identities, ranging from 88.4 to 99.3% and from 95.4 to 99.7%, respectively. Among the five isolates, the above were 93.7–99.3% and 97.7–99.7%, respectively (including CAstV/CK/CH/HBBD/6, HBLF/41, HBBD/52, HBSJ/80, and HBCZ/117). The six isolates were clustered with the reference sequences of CAstV from India, Canada, the UK, and the US (Figure 3). The six isolates were clustered with the reference sequence of strain 4175 (JF832365), with a bootstrap value of 100 in the common ancestral line, which revealed that the six isolates should be classified into the Bii serotype.

PCR products and corresponding nt sequences (2022–2049 bp) of a complete region of the ORF2 gene of eight ANVs were obtained through RT-PCR. The nt and predicted amino acid sequences of the ANV ORF2 gene were found to exhibit low sequence identities from 61.4 to 98.0% and 60.4 to 98.7%, respectively. A second group of eight ANV isolates (ANV/CK/CH/HBBD/32, HBCZ/69, HBBD/77, HBCZ/106, HECZ/117, HESJZ/126, HESJZ/127, and HBBD/134) were clustered with nine reference sequences of ANV isolates (Appendix A) from the UK, Brazil, Australia, China, and Japan (Figure 3). Isolate ANV/CK/CH/HBCZ/106 was clustered with an isolate of ANV from the UK. In addition, isolate ANV/CK/CH/HBBD/32 was clustered in a different group, together with the Brazilian isolates, with a bootstrap value of 100 in the common ancestral line. Isolates HBBD/77 and HBBD/134 were clustered with an isolate of ANV from China with a bootstrap value of 100 in the common ancestor line. Furthermore, isolates ANV/CK/CH/HBCZ/69, HESJZ/127, HECZ/117, and HESJZ/126 were clustered with the reference sequences of two ANV from Brazilian isolates. The eight ANV isolates pertained to multiple genotypes (Figure 3).

## 4. Discussion

Enteric viruses of poultry can trigger significant economic losses, since they may decrease bird weight gain, increase morbidity and mortality, and increase production costs from poor feed conversions [4,6,23,24]. Various results have been achieved in the investigation of seven enteric viruses in different countries, including profiles of virus combinations, and the main enteric viruses. In the positive samples, 11 profiles of virus combinations were detected (Table 5), 73.1% of samples were subjected to 106 single infections, and multiple infections were detected among the remaining 26.9% of the samples. ChPV was found as the main virus either in single infection or multiple infections (Table 4 and Table 7). The Indian research regarding ANV, CAstV, and reovirus in broiler chickens reported that 53.80% of samples tested positive for a single virus, 40.00% for two viruses, and the remaining samples were negative for all three viruses tested. Moreover, none of the cases tested positive for reovirus [24]. The results of the above study are consistent with the results of this paper, which may be correlated with their geographical proximity. As revealed by the molecular examination of the seven enteric viruses of commercial chicken flocks in South Korea between 2010 and 2012, among 28 positive samples, there were 50% single and multiple infections, respectively, forming 15 profiles of virus combinations; the most common pathogen found was ANV, and the least common was FAdV [6]. The molecular examination of enteric virus of commercial chicken flocks in Brazil from 2010 to 2017 suggested that 25 profiles of virus combinations were detected in 270 samples, while single infections were detected in 86.3% of samples, and multiple infections were reported in the remaining 13.7%. IBV was the most common virus detected, and ARe was the least common [7]. The above results are consistent with the detection results regarding commercial chicken in Brazil from 2008 to 2010, namely the viruses most frequently detected, either alone or in concomitance with other viruses, were IBV, ANV, rotavirus, and CAstV [3]. The main pathogens detected by the above researchers revealed that the abundance of common enteric viruses varied significantly among different regions.

The correlation of enteric virus with the age of birds suggested that the viruses occurred at different stages in the chickens (Table 6 and Table 7), which may be conducive to the control of virus infections. In the results of this paper, ANV and CAstV were primarily present in broilers and the first 2–11 weeks of age of the pullet/layer hens. Considerable ChPV was detected in all age groups of broilers and hens. Impacted by the limitation of the experimental materials, broilers under 15 days in age and pullets of 1 week in age were not investigated in this study. As revealed by the results of this paper (Table 6 and Table 7), enteric viral infections largely affected young birds, most likely because the mucosal barrier and defense system of young birds have not yet fully developed, and the birds are easily invaded by viruses. Moreover, the rapid proliferation of intestinal cells in young birds would create a suitable environment for proliferating viruses, which may account for the high positive rate of parvovirus in young birds [25]. Avian astroviruses, both CAstV and ANV, were mostly detected in the samples from young birds, despite five positive samples being found in breeders older than 22 weeks (Table 6). The above result is consistent with the conclusions of David et al. (2018), namely that astrovirus infections usually affect young birds, but can also be detected in older birds [7]. Parvovirus affects birds of virtually all ages, as confirmed by the results of this paper. This finding differs from existing reports [7,26] which suggest that parvovirus infections affect young birds in their first 4 weeks. The reason for this is that, in this paper, parvovirus was detected not only in the 11 infected samples from broilers younger than 28 days, but also in a considerable number of older broilers and pullet/layer hens, which may be related to the local epidemic environment of parvovirus. As revealed by the results of Zhang et al. (2020), ChPV and TuPV are widely distributed in commercial fowl in some areas of China (e.g., Guangxi). The highest frequency of ChPV positive samples in chickens was 64.18% in broiler chickens, compared with 38.75% in breeder chickens and 38.89% in layer hens [14]. Moreover, TuPV was detected in 83.33% of the samples. The data of this study suggest that ChPV is widely distributed in commercial poultry flocks in Hebei Province, China (Table 4 and Table 7). The age of the birds relating to the infection of FAdV-I and ARoV could not be observed in this paper, since the positive samples were significantly low (Table 7).

The clinical symptoms were classified, and the results of the Fisher’s exact analysis suggested that there was no significant difference in clinical signs between the two types of birds. The correlation between enteric viruses and clinical symptoms is reported as follows. ChPV has also been identified in a high proportion of chicken flocks suffering from enteritis [6]. Furthermore, the results of our previous study showed that the ChPV positive rate was 48.3% in 151 chicken small intestine samples with enteritis. Koo, Lee, et al. (2013) reported that other pathogens (e.g., *Escherichia coli, Salmonella* spp.) were detected in most of chicken flocks positive for enteric viruses [6]. Additionally, co-infections of CAstV and ANV may affect visceral gout disease severity [21], but research regarding this was conducted in this paper. The role of each enteric virus in chick diseases remains unclear.

The above enteric viruses may play a role in the etiology of enteric diseases in poultry [23,27]. Parvovirus sequences were characterized in accordance with their positions in the phylogenetic trees, which indicated the nucleotide phylogenetic tree of complete parvovirus sequences clustered in a similar manner as the phylogenetic tree based on VP1 sequences (Figure 1 and Appendix A). However, the above two phylogenetic trees clustered differently from the phylogenetic tree of NS (Appendix A), since the isolate TuPV/CK/CH/HB/22 was clustered in TuPV. The NS gene was found as the basis of parvovirus classification by the International Committee on Taxonomy of Viruses [28]. All of the above seven isolates were isolated from chickens, whereas TuPV/CK/CH/HB/22 was termed based on NS classification. The results of the phylogenetic analysis that compared NS gene sequences suggested a strong similarity between the chicken and turkey parvoviruses. Notably, most of the TuPV and ChPV strains formed distinct phylogenetic groups, which revealed that the above viruses may have diverged from a common ancestor, and have subsequently undergone a host-specific adaptation. Consequently, the chicken and turkey parvoviruses were highly adapted to their respective host species. Zsak et al. (2015) found that the viral structural VP1 gene primarily accounted for this host specificity [23]. Based on the strong species specificity, potential heterologous, cross-species infection with parvoviruses would not play a vital role in the epidemiology among species. However, in the present paper, recombination was found (Figure 2), which is consistent with the structural characteristics of parvovirus, i.e., a single-stranded DNA virus. The genetic evolutionary rates of single-stranded DNA viruses (e.g., parvoviruses) are higher than those of double-stranded DNA viruses [25,29], and recombination events have been identified in parvoviruses, which were also verified here. According to some gene sequences, both NS and VP1/VP2 junctions exhibited high diversity [30,31]. The phylogenetic tree parvoviruses isolates in this paper were found to have a close phylogenetic relationship with isolates from India, the US, Poland, South Korea, Brazil, and Hungary, thus further confirming the distribution of parvoviruses around the world [2,6,7].

Both CAstV and ANV were identified in a high proportion of chicken flocks in this study, especially in broilers, and coinfection was also detected (Table 5). Their genetic organization consists of three open reading frames (ORFs): ORF1a, ORF1b, and ORF2, among which ORF2 encodes the capsid protein. The capsid gene sequence of astroviruses is highly variable, and is significantly correlated with antigenicity [32,33]. The serogroup was further supported in the genotyping study, in which the strains were clustered into CAstV groups A and B based on a lower level of shared amino acid identity across ORF2 (38–40%) [33]. Furthermore, group A was assigned to three subgroups (i.e., Ai, Aii, and Aiii), while group B fell into four subgroups (i.e., Bi, Bii, Biii, and Biv) [21]. As compared with the known subgroups [20,21], the six CAstV isolates identified in this paper were classified into Bii, based on phylogenetic analysis of a complete sequence of ORF2. Smyth (2017) found that the subgroups of Biii and Biv were associated with specific broiler chick diseases, namely kidney disease with visceral gout, and white chicks, respectively [21]. The hazards of subgroups of Bii alone have not been elucidated. It is of significance to classify CAstV according to ORF2 since it is where the most hypervariable regions associated with antigenicity are located. As revealed by the analysis of the ORF2 of eight ANV isolates, two strains were significantly correlated with Chinese strains, while the other six strains were closely correlated with strains from other countries (Figure 3). This revealed that there was no direct correlation between the homology of different strains and the distance of their genetic relationship and the region. The eight strains pertained to multiple genotypes, thus suggesting that there were multiple ANV infections in chickens in China. The occurrence of mixed infection verified the possibility for the genome evolution of ANV and recombination between different strains [19].

Because microbial colonization occurs simultaneously with the development of intestinal tissue of young animals [34], the interaction between the microbial environment and mucosal host cells affects the function of the intestinal system, including nutrient absorption, maintenance of intestinal barrier integrity, and immune homeostasis. Now studies have shown that enteric viruses disrupt the integrity of the intestinal barrier and immune homeostasis. Therefore, our results have certain practical significance in clarifying the distribution of enteric virus. Furthermore, ORF2 is an important component of capsid protein. The complete sequence of ORF2 obtained in this report can be analyzed bioinformatically, and further provide a theoretical basis for the preparation of vaccines and the development of detection kits.

Chicken enteric virus, a crucial factor posing a hazard to the development of chicken breeding, is easily ignored. In this paper, the prevalence and genetic evolution of chicken enteric virus in Hebei Province from 2019 to 2021 were investigated. The results of this paper lay a theoretical foundation for the prevention and control of chicken enteric virus disease in the future.

## 5. Conclusions

In the investigation of seven enteric viruses in Hebei Province, China, ChPV was shown to be the most common virus detected, followed by ANV and CAstV, yet neither IBV nor ARe were detected in the samples. In the broilers, dual virus was found to be the main infection type, while in the pullet/layer hens, single infection was found as the main infection type. As revealed by the phylogenetic trees, there were six isolates corresponding to ChPV, while one isolate may have undergone recombination events. The six CAstV isolates were classified into the Bii serotype, and the eight ANV isolates pertained to multiple genotypes.

## Figures and Tables

**Figure 1 animals-12-02873-f001:**
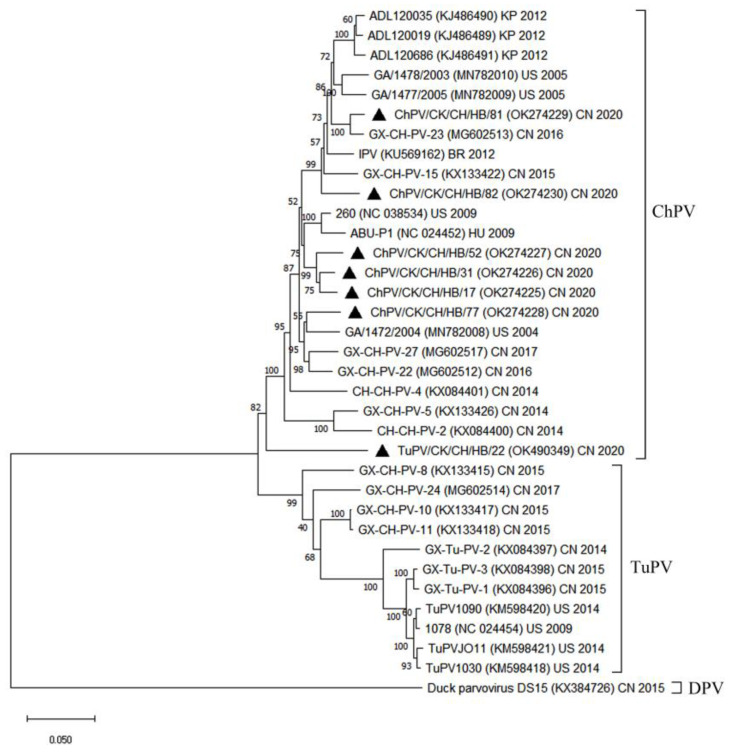
Phylogenetic tree based on the complete sequence gene of parvovirus. Molecular Evolutionary Genetics Analysis version 11 was used to perform the phylogenetic tree reconstruction, using the neighbor-joining algorithm with 1000 bootstrap replicates. The tree was drawn to scale, with branch lengths in the same units as those of the evolutionary distances used to infer the phylogenetic tree. Accession numbers are shown in parentheses. Strains with ▲ represent PV sequenced in this paper. The sequence of duck parvovirus was used as the out-group control. CN, China; US, United States; HU, Hungary; BR, Brazil; KR, South Korea.

**Figure 2 animals-12-02873-f002:**
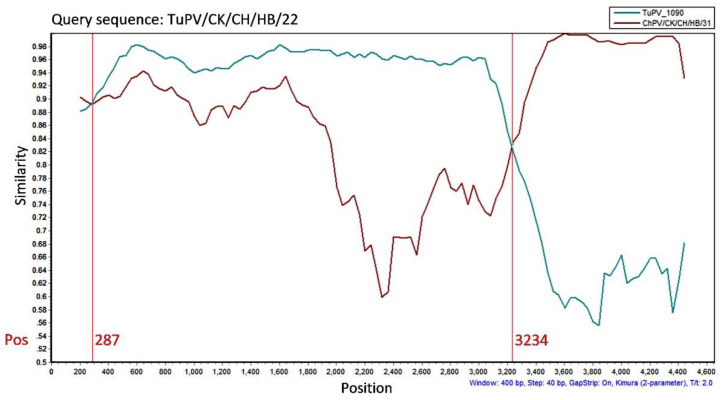
SimPlot analysis of recombinant TuPV/CK/CH/HB/22 sequence for confirming recombination was performed using the Simplot software program v3.5.1. The analysis considered different parent sequences (strain TuPV-1090, TuPV/CK/CH/HB/31) plotted in a graph considering the vertical-axis percentage of permuted trees, and on the horizontal axis is the position on the genome of the query sequence.

**Figure 3 animals-12-02873-f003:**
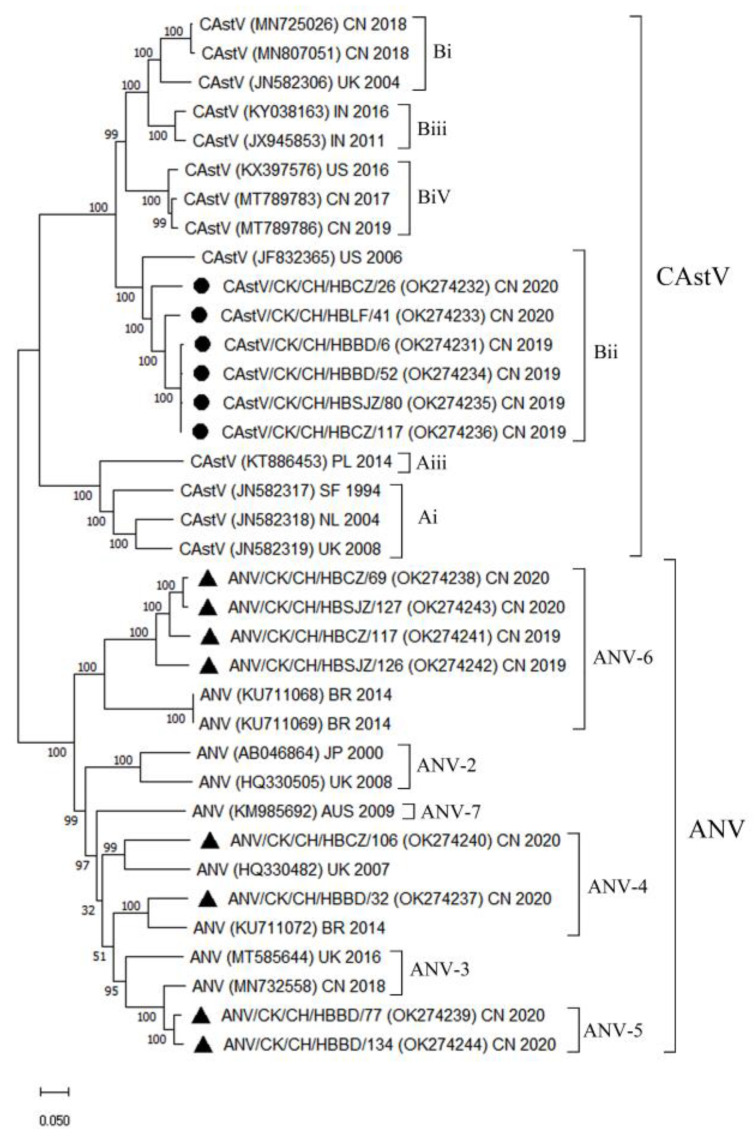
Phylogenetic tree based on analysis of open reading frame 2 (ORF2) complete gene of chicken astrovirus (CAstV) strains and avian nephritis virus (ANV) strains. Molecular Genetics Evolutionary Analysis version 11 was used for phylogenetic tree reconstruction using the neighbor-joining algorithm with 1000 bootstrap replicates. These included sequences of different genotypes of CAstV and ANV according to ORF2 analysis described by references [20,21,22]. The tree was drawn to scale, with branch lengths in the same units as those of the evolutionary distances used to infer the phylogenetic tree. The accession numbers are expressed in parentheses. Strains with ● and ▲ represent the CAstV and ANV identified in this paper, respectively. CN, China; UK, United Kingdom; IN, India; US, United States; PL, Poland; SF, South Africa; NL, Netherlands; JP, Japan; AUS, Australia; BR, Brazil.

**Table 1 animals-12-02873-t001:** Primers sequences, gene target, amplicon size, and the corresponding references that were used to screen for the enteric viruses by RT-PCR and PCR reactions.

Virus	Primer Name	Primer Sequences (5′ to 3′)	Target Gene	Amplicon Size (bp)	Annealing Temp (°C)	References
ANV	ANV-F ANV-R	RCTRGGCGCCTCTTTTGAYW CRTTKCCCKGTAGTCTYTGA	ORF1b	473	52	[9]
CAstV	CAstV-F CAstV-R	GAYCAKCGAATGCGRAGGTTR TCRGTGGGAGTGGGKAGTCTRC	ORF1b	362	52	[9]
ARoV	ARoV-F ARoV-R	GTGCGGAAAGATGGAGAAC GTTGGGGTACCAGGGATTAA	NSP4	630	52	[9]
ARe	ARe-F ARe-R	GGTGCGACTGCTGTATTTGGTAAC AATGGAACGATAGCGTGTGGG	S1	532	57	[16]
IBV	IBV-F IBV-R	CCTAAGAACGGTTGGAAT TACTCTCTACACACACAC	M	739	54	[15]
FAdV-I	FAdV-I-F FAdV-I-R	TGCTCGTTGTGGATGGTGAA CTCCGTGTTGGGCTGGTC	Polymerase	594	57	[18]
ChPV	ChPV-F ChPV-R	TAACGATATCACTCAAGTTTC GCGCTTGCGGTGAAGTCTGGC	NS	552	53	[17]

Abbreviations: ANV, avian nephritis virus; CAstV, chicken astrovirus; ARe, avian reovirus; ARoV, avian rotavirus; IBV, infectious bronchitis virus; FAdV-I, fowl adenovirus of group-I; ChPV, chicken parvovirus; ORF1b, open reading frame 1b; S1, spike 1 gene; NSP4, nonstructural protein 4; M, membrane protein gene; NS, nonstructural gene.

**Table 2 animals-12-02873-t002:** Primers used in this study for polymerase chain reaction amplification of the complete genome of chicken parvovirus (ChPV) and turkey parvovirus (TuPV).

Primer	Primer Sequences (5′ to 3′)	Amplicon Size (bp)	Annealing Temperature (°C)	Extension Time (s)	Primer Location (nt)	References
PV-AF	CAATCCGCACGTTGATTCGG	303	57	30	ChPV/TuPV 366–668	[6]
PV-AR	GGCGTGTTTCGCCAATTGAA					
PV-BF	TATGCCTGTATTCGAGTGTG	1559	52	100	ChPV 485–2043	
PV-BR	CTTTCAGGAGACTATCCACG					
PV-CF	TTCCACGGACCAGCCAATAC	1360	52	100	ChPV 1876–3915	
PV-CR	CATATCCCTGCTGGGTCGTC					
PV-DF	CTCGAGTGGTACAATGGGGG	569	55	45	ChPV 2613–3181	
PV-DR	AGCGTTTGCGTTCAGCTTTT					
PV-EF	GCTTCTATAGGCACACAATG	895	50	50	ChPV 2981- 3875	
PV-ER	CCTGTCAAGTCGTTAGAGTA					
PV-FF	ACCGGTAATTGGAATTGTGA	1448	52	100	ChPV 3520–4967	
PV-FR	ACATTGACCTGGTATTGACC					
PV-GF	GAACCACTCAACACTCACAG	354	53	30	ChPV/TuPV 4855–5208	
PV-GR	CATATGCATAGTCACGCCTT					
PV-HF	CTGCTGAGCTGGTAAGATGG	2007	64	130	TuPV 395–2401	[14]
PV-HR	TTTGCGTTGCGGTGAAGTCTGGCTCG					
PV-IF	TTCTAATAACGATATCACTCAAGTTTC	1838	48	60	TuPV 1841–3678	
PV-IR	GTATTGKGTYTGGTTTTCAG					
PV-JF	CAAGCCGCCATTGTGTTTGT	1451	57	100	TuPV 3575–5025	
PV-JR	TTAATTGGTYYKCGGYRCSCG					

**Table 3 animals-12-02873-t003:** RT-PCR amplification primers of complete open reading frame 2 (ORF2) gene of avian nephritis virus (ANV) and chicken astrovirus (CAstV).

Primer Name	Primer Sequences (5′ to 3′)	Amplicon Size (bp)	Annealing Temperature (°C)	Extension Time (s)	References
ANV-ORF2-F	ACCTTGAATCCCTGTGGGGCA	2500	55	130	[19]
ANV-ORF2-R	AAAAGTTAGCCAATTCAAAATTAATTC				
CAstV-ORF2-F	CGGGATCCATGGCCGATAAGGCTGGGCCG	2214	63	130	This research
CAstV-ORF2-R	CGGAATTCCTACTCGGCGTGGCCGCG				

**Table 4 animals-12-02873-t004:** Frequencies of viruses and samples in single and multiple viral infections diagnosed ^1^.

Item	All	Infection Type			χ^2^	*p*-Value ^4^
		Single Infection	Multiple Infection (2, 3, 4 Viruses)			
Virus ^2^						
ChPV	100	69 (69.0%) ^3^	31 (31.0%)	(25, 5, 1)	15.494	0.000 **
ANV	44	17 (38.6%)	27 (61.4%)	(21, 5, 1)	10.040	0.002 **
CAstV	41	18 (43.9%)	23 (56.1%)	(18, 4, 1)	6.769	0.009 **
FAdV-1	5	2 (40.0%)	3 (60.0%)	(2, 1, 0)	-	0.657
ARoV	1	0 (0.0%)	1 (100%)	(0, 0, 1)	-	0.445
Total	191	106	85	(66, 15, 4)		
Positive samples, *n*	145	106	39	(33, 5, 1)		
Positive samples/145		73.1%	26.9%	(22.8%, 3.4%, 0.7%)		

^1^ Abbreviations: ANV, avian nephritis virus; CAstV, chicken astrovirus; ARe, avian reovirus; ARoV, avian rotavirus; IBV, infectious bronchitis virus; FAdV-I, fowl adenovirus of group-I; ChPV, chicken parvovirus. ^2^ IBV and ARe were not detected in any samples. ^3^ Positive/total number of each enteric virus detected. ^4^ Comparison between two types of infections (single and multiple infection) by chi-squared or Fisher’s exact test. ** Indicates extremely significant differences.

**Table 5 animals-12-02873-t005:** Enteric virus detection patterns from 145 positive samples in the small intestines from chicken commercial flocks.

Pattern	Virus Examined in 145 Samples							Number of Samples and the %
	FAdV-I	ChPV	CAstV	ANV	IBV	ARe	ARoV	
1	+	+		+				4 (2.8%)
2		+	+	+				1 (0.7%)
3		+	+	+			+	1 (0.7%)
4		+		+				14 (9.7%)
5	+	+						2 (1.4%)
6		+	+					10 (6.9%)
7			+	+				7 (4.8%)
8		+						69 (47.6%)
9				+				17 (11.7%)
10			+					18 (12.4%)
11	+							2 (1.4%)

Abbreviations: ANV, avian nephritis virus; CAstV, chicken astrovirus; ARe, avian reovirus; ARoV, avian rotavirus; IBV, infectious bronchitis virus; FAdV-I, fowl adenovirus of group-I; ChPV, chicken parvovirus. + indicate that the sample showed positive result in the PCR assay, while blank indicates negative.

**Table 6 animals-12-02873-t006:** The number and positive rate of viral infections according to the ages of layer and broilers flocks.

Pullet/Layer Hens			Broilers		
Age of Flocks	Number of Positive Samples	Positive Rate for Each Week of Age	Age of Flocks	Number of Positive Samples	Positive Rate for Each Day of Age
2–6 weeks	34	87.2% (34/39) ^1^	15–21 days	3	100.0% (3/3) ^1^
7–11 weeks	31	67.4% (31/46)	22–28 days	9	75.0% (9/12)
12–16 weeks	14	82.4% (14/17)	29–35 days	9	75.0% (9/12)
17–21 weeks	29	65.9% (29/44)			
22–26 weeks	11	50.0% (11/22)			
>26 weeks	5	29.4% (5/17)			
Total	124	67.0% (124/185)		21	77.8% (21/27)

^1^ Enteric virus positive samples/total number of samples detected for enteric virus in each group.

**Table 7 animals-12-02873-t007:** Frequency of viruses affecting birds according to bird age ^1^.

Chicken Line	Age of Birds	FAdV-1	ChPV	CAstV	ANV	ARoV	ARe	IBV
Broilers	15–21 days (*n* = 3)	0	3	0	3	0	0	0
	22–28 days (*n* = 9)	0	8	1	5	0	0	0
	29–35 days (*n* = 9)	1	7	4	2	0	0	0
Pullet/layer hens	2–6 weeks (*n* = 34)	0	19	12	15	0	0	0
	7–11 weeks (*n* = 31)	0	18	10	12	1	0	0
	12–16 weeks (*n* = 14)	0	11	5	0	0	0	0
	17–21 weeks (*n* = 29)	3	23	6	5	0	0	0
	22–26 weeks (*n* = 11)	0	9	2	1	0	0	0
	>26 weeks (*n* = 5)	1	2	1	1	0	0	0
Total	(*n* = 145)	5	100	41	44	1	0	0

^1.^ Abbreviations: ANV, avian nephritis virus; CAstV, chicken astrovirus; ARe, avian reovirus; ARoV, avian rotavirus; IBV, infectious bronchitis virus; FAdV-I, fowl adenovirus of group-I; ChPV, chicken parvovirus.

**Table 8 animals-12-02873-t008:** Positive samples with single and multiple viral infections and clinical signs according to bird type.

Item	All, *n* = 145	Chicken Line		χ^2^	*p*-Value ^3^
		Broilers, *n* = 21	Pullet/Layer Hens, *n* = 124		
Positive samples of infections					
One virus	106 (73.1%) ^1^	9 (42.86%)	97 (78.22%)	10.423	0.001 **
Two viruses	33 (22.8%)	11 (52.38%)	22 (17.7%)	-	0.001 **
Three viruses	5 (3.5%)	1 (4.76%)	4 (3.2%)	-	0.548
Four viruses	1 (0.7%)	0 (0.0%)	1 (0.8%)	-	1.000
Clinical signs					
Respiratory problems	16 (11.0%)	3 (14.3%)	13 (10.5%)	-	0.705
Digestive problems ^2^	54 (37.2%)	9 (42.9%)	45 (36.3%)	0.331	0.565
Stunting	33 (22.8%)	7 (33.3%)	26 (21.0%)	-	0.260
Not reported	42 (29.0%)	2 (9.5%)	40 (32.3%)	-	-

^1^ Number of positive samples, % of positive samples independently of chicken line. ^2^ Digestive problems were defined as enteritis, diarrhea, decreased feed absorption, and indigestion of feed. ^3^ Comparison between two chicken lines (broilers and pullet/layer hens) by Chi-squared or Fisher’s exact test. ** Means extremely significant differences.

## Data Availability

All relevant data are within the manuscript and its Appendix A. Further inquiries can be directed to the corresponding authors.

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
