# Peer review of "Detection and Molecular Characterization of Enteric Viruses in Poultry Flocks in Hebei Province, China"

_animals, 2022, doi:10.3390/ani12202873_

Round 1
Reviewer 1 Report
The manuscript " Detection and Molecular Characterization of Enteric Viruses in Poultry Flocks in Hebei Province, China", is generally well-addressed and well-written however; I have some comments:
Line 13: please delete “so on” or replace with etc.
Line 43: The introduction is very concise and need to be revised. It need to be improved with details about the enteritis as important diseases in poultry industry and caused viruses. Also I suggest to add few lines about the situation and origin of these viruses in china (molecular epidemiology) then, talk about the recent finding characterization of enteric viruses in china as well as other countries.
Line 81: The samples were processed and analyzed......: I suggest this details to be under samples preparation.
Line 165: correct the reference “endnote error”, same at Lines 409, 422 & 448
Line 181:in the sentence “virus detected corresponded to ChPV, showing a 52.4% (100/191) occurrence .....”please provide a column in table 4 for the virus occurrence percent to be clear for readers. You just add the number of infections (e.g 100/191).
Line 201: Table 4: please revised it: please abbreviation of X2, also, in figure legend this “comparison between two types of infection” is not indicated in the table.
Table 5: correct the column of “Number of samples and the %”.
Line 226: the sentence “rate of viral infections, which was .....”The rate of infections in text is different from that in the table 6.
Table 6: positive rate for each week: provide in the legend the calculation of it to be clear. For example: 87.2% (34/39), this is the number of positive samples divide by 39, what 39 is ?
Table 8: in the legend: 3 is “Comparison between two chicken line ...” I cannot find 3 in the table, also please provide the abbreviation of X2
L384: In the discussion the words " in this paper" is extensive repeated, please revise it.
Line 406: Zhang et al. 405 (2020)13, correct reference 13 here
Reviewer 2 Report
The work presented for review concerns a very interesting issue, which is contamination of farms breeding chickens with viruses.
As always, in large production, threats that reduce bird welfare can cause significant economic losses, and as viruses have the ability to change quickly, the threat seems to be even greater.
The work is prepared in a careful manner, the experiment is planned in accordance with the intended purpose of detecting and characterizing viruses in the poultry house environment.
Both the description of the experiment and the discussion of the results do not raise any objections, no self-citations were detected, and the cited literature is adequate to the topic and, to a certain extent, new.
Remarks:
1. Figure 2 is not legible. Please correct the sharpness of the chart.
2. I am aware that the aim of the work was screening and characterization of identified viruses, but at the end of the discussion the authors write: "The results of this paper lay a theoretical foundation for the prevention and control of chicken enteric virus disease in the future". Actually nowhere in the text of the manuscript have I found what this prevention should consist of? Should any new, specific methods of prevention be developed for these viruses and the dangers they entail? I recommend the Authors to expand or add a paragraph in the Introduction or Disscusion section, about the preventive measures applied or proposed on poultry farms, so that the reviewed article has at least minimal practical relevance.
